# A New Picking Pattern of a Flexible Three-Fingered End-Effector for Apple Harvesting Robot

**Wei Ji \*, Guozhi He, Bo Xu, Hongwei Zhang and Xiaowei Yu**

School of Electrical and Information Engineering, Jiangsu University, Zhenjiang 212013, China;
hegz3722@163.com (G.H.); xubo@ujs.edu.cn (B.X.); zhanghongwei991016@163.com (H.Z.);
18879209223@163.com (X.Y.)
**\*** Correspondence: jiwei@ujs.edu.cn

**Abstract:** During the picking process of the apple harvesting robot, the attitude of the end effector holding the apple and the movement method of separating the apple directly affect the success rate of picking. In order to improve the stability of the picking process, reduce the gripping force, and avoid apple dislodgement and damage, this work studies the new apple-picking pattern of the flexible three-fingered end-effector based on the analysis of the existing apple-picking pattern. First, two new three-finger grasping postures for wrapping the apple horizontally and vertically on the inside of the fingers are proposed, and a new method of separating the stem with a circular-pull-down motion of the end-effector picking the apple is designed. Then, the pressure on the apple under different picking patterns was analyzed, and a branch–stem–apple simulation model was established. Combining the constraint conditions such as the angle between the apple stem and the vertical direction, the movement speed, the root impulse, and so on, the optimal angle of apple circular movement and the force required to realize the movement are obtained through dynamic simulation experiments. Finally, the experiments of apple picking patterns were carried out with the flexible three-fingered end-effector. The experiment shows that the best angle for apple picking is 15°~20° using the circular-pull-down movement separation method. In terms of average grasping force peaks and pressures, the combination of the vertical holding posture of the inner finger and the circular-pull-down movement separation method is the best picking pattern. In this pattern, the average peak exerts force on the inner side of a single finger is about 8.52 N, and the pressure is about 20.9 KPa.

**Keywords:** harvesting robot; end-effector; picking pattern; grasping posture; separation method





## 1. Introduction

Fresh apples are an essential part of the human diet. With the expansion of orchards around the world, the problems of time-consuming apple picking, high labor costs, and the inability to pick apples in bad weather are beginning to emerge. Nowadays, computer applications, artificial intelligence, and automatic control technologies are maturing, and apple harvesting robots have become a new trend to replace manual apple picking [1]. The gripper is a key component of the harvesting robot that is in direct contact with the fruit, and its performance directly affects the picking effect of the harvesting robot.

Over the past three decades, researchers have worked on designing different grippers to find ways to enable efficient and stable apple picking [2]. A novel agricultural picking robot has been designed with an adsorption-type tubular end-effector that outputs sufficient vacuum flow to attract and pick apples within a certain distance [3]. However, the picking effectiveness was limited by the size of the tubular end-effector aperture. Based on the traditional rigid two-finger type gripper, a flexible structure has been added to the finger end side to reduce the damage rate when picking apples [4]. In order to improve the suppleness of the picking process, a flexible finger is used instead of the previously rigid,

mechanically structured fingers [5,6]. The conventional two-finger type gripper has a single grasping posture, which makes it difficult to achieve stable and efficient picking in complex environments. The three-fingered gripper has the characteristics of strong grasping force, flexible contact, and strong adaptability [7,8]. In complex environments, it can accomplish the grasping of objects in different postures. This shows that soft-finger grippers are the trend of development due to their good encapsulation.

During the grasping of apples by the gripper, the operational forces that achieve the desired movement of the apples and the internal forces that maintain the dynamic stability of the apples constitute the grasping forces for apple picking. The internal forces influence the stability of the apple grasping process and the degree of crushing. To maintain a stable grasping action, the internal force should be within a reasonable range, and the gripper-apple surface contact should be in a static frictional constraint [9]. Excessive grasping force and detachment force can lead to damage to the apple skin and cause unnecessary waste [10]. The stresses on apples under static loading were investigated using ultrasonic techniques, and the maximum gripping force threshold for apple damage was determined. As long as the gripping force on the apple is within the allowable threshold, damage to the apple to an unacceptable degree can be avoided [11]. By building a collision prediction model, it was found that the degree of impact caused by static or dynamic gripping affects the degree of skin abrasion during apple gripping by a gripper, and it was also found that factors such as fruit ripeness, fruit temperature, radius of curvature, and acoustic hardness determine the susceptibility to fruit bruising [12]. Three gripping postures and two separation methods based on the three-finger apple-picking model are proposed by observing the manual apple-picking posture [13]. Based on the force, damage, and swing amplitude of the apple in the picking experiment, the picking pattern closest to manual picking was derived: the separation methods in which the fingertips are held parallel to the calyx axis of the apple stem, causing the apple to make a bending and pulling down motion. In contrast, Four easier-to-achieve separation methods were designed based on the three fingertip grasping positions: vertical downward pull, horizontal pull, rotational pull, and rotational horizontal pull [14]. Comparative experiments show that the rotary horizontal pull is superior to the other three separation methods. Combining these grasp postures and separation methods results in a picking pattern that enables fast picking of apples but makes it difficult to ensure stable apple movement during picking and avoid damage caused by movement. The effect of different separation motions on the apple de-stemming process can be effectively analyzed by building an apple branch–stem–fruit model. The stability of the transportation process is observed while ensuring successful branch–stem–fruit model breakage [15,16]. With the use of three-finger type soft-finger grippers [17], an in-finger grasp stance has also become feasible. It is, therefore, necessary to plan a better grasp posture and apple-picking action for the three-fingered gripper on the apple surface to ensure stability and minimize force peaks and potential damage during the picking process.

Based on the above analysis, in order to solve the instability and reduce the force on apples during the grabbing process, this paper designs two new gripper finger inside-out apple grasping postures and a new circular-pull-down separation method to release apples based on the analysis of existing apple picking patterns. Then, the optimal state conditions for circular pull-down separation of apples were determined by performing dynamic simulations and designing apple-picking experiments. Finally, real apple-picking experiments were conducted to demonstrate the superiority of this method, as shown in Figure 1.

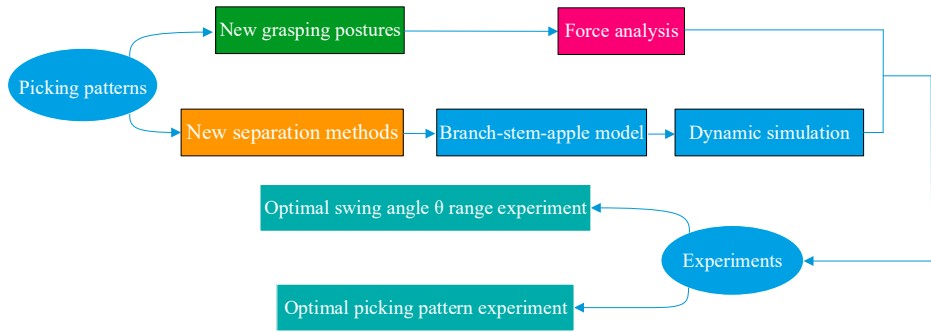

**Figure 1.** Flowchart of the research in this paper.

## 2. Materials and Methods

*2.1. Flexible Three-Fingered End-Effector Picking Pattern Design*

2.1.1. Grasp Posture Design of Gripper

For the study of flexible three-fingered end-effector grasp postures, two three-finger fingertip grasp postures were designed by Li et al. [13], as shown in Figure 2a,b. They are Posture 1: fingertip horizontal centripetal grasp (two adjacent fingers placed at 120° apart near the equatorial surface of the apple) and Posture 2: fingertip parallel grasp on the calyx of the apple stem (two fingers placed parallel to the side of the stem and one finger placed at the base of the apple). The apple-separation experiments were carried out for both grasping postures, ensuring that the separation methods and environmental factors were the same. The experiments showed that the grasping force, grasping pressure, and damage rate of apples picked using grasp Posture 2 were lower than those of grasp Posture 1 and that the results were closer to those of manual picking. However, this method only considers the fingertip grasp posture and ignores the inner finger grasp posture.

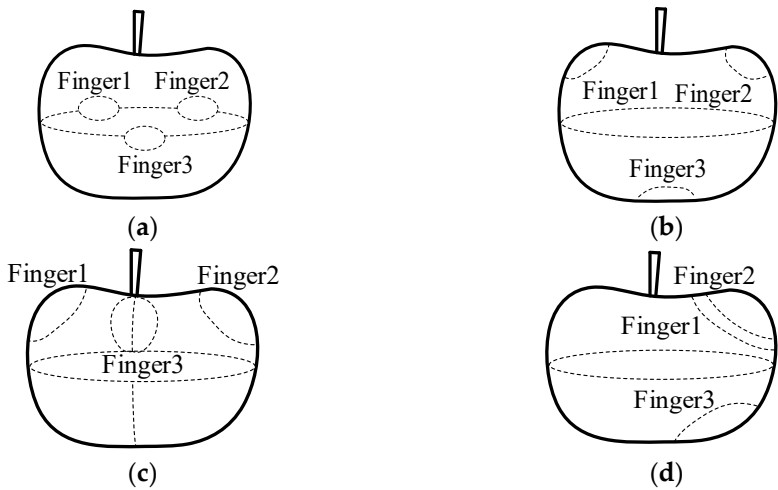

**Figure 2.** Four three-finger grasp postures. (**a**,**b**) The fingertip grasp posture proposed in [13]; (**c**,**d**) the new posture of the medial finger grasp proposed by this paper. (**a**) Grasp Posture 1 (proposed in [13]), (**b**) Grasp Posture 2 (proposed in [13]), (**c**) Grasp Posture 3 (proposed), (**d**) Grasp Posture 4 (proposed). The dashed portions indicate the cross section of the apple and the contact area with the finger.

In this study, two medial finger grasp postures are proposed, as shown in Figure 2c,d. These are the Posture 3 medial finger horizontal centripetal grasp (three fingers wrapped around the apple, fingertips placed at 120° apart on the calyx side, palm resting on the bottom of the apple) and Posture 4 medial finger parallel to the calyx axis grasp (fingertips oriented in a vertical calyx direction, grasping the waist side of the apple, palm resting on the other waist side). When picking apples using the inner finger grasp posture, the soft finger has a larger contact area with the apple than in the previous two fingertip grasp

postures. At the same time, the cushioned area in the palm of the gripper shares a certain amount of force and reduces the pressure on the skin of the apple.

When picking apples using the inner finger grasp posture, the soft finger has a larger contact area with the apple. At the same time, the cushioned area in the palm of the gripper shares a certain amount of force and reduces the pressure on the skin of the apple.

### 2.1.2. Stem Separation Method Design of Gripper

In terms of research to achieve fruit stem separation, a flexible three-finger end-effector bending and pulling motion method was designed to achieve fruit stem separation (defined as separation method 1: shown in Figure 3a) by Li et al. [13]. The gripper drags the apple by applying a pulling force in the direction of the rootstock growth, causing the apple to move in a pendulum motion perpendicular to the rootstock. During the picking process, the stem-branch node is bent by force, and the combination of the vertical direction pull, and the pendulum motion results in fruit separation. A rotary-horizontal pulling motion separation method was used (defined as separation method 2: shown in Figure 3b) by Fan et al. [14]. The apple is separated in a rotary motion separation method with the core as the center of the circle and the rootstock plane, while the gripper pulls the apple horizontally to complete apple picking.

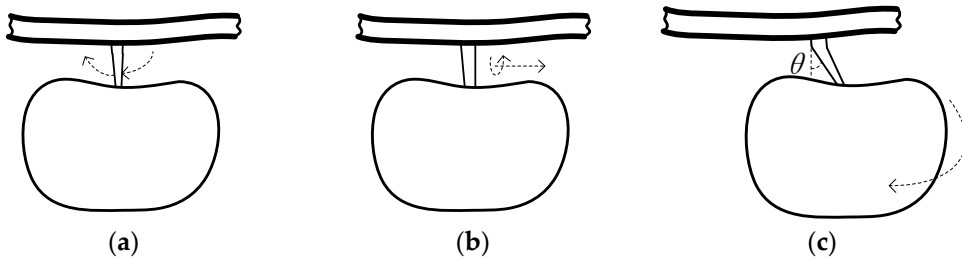

|     |     |     |
| :-: | :-: | :-: |
| (**a**) | (**b**) | (**c**) |

**Figure 3.** Three three-finger fruiting pedicel separation motions. (**a**) Separation method 1: the curved-pulling motion separation method proposed in [13]; (**b**) separation method 2: the rotary-horizontal pulling motion separation method proposed in [14]; (**c**) separation method 3: the circular-pull-down motion separation method proposed by this paper. (**a**) Separation method 1 [13], (**b**) separation method 2 [14], (**c**) separation method 3 [proposed]. The dashed portion of the line indicates the direction of apple undergoing stem separation movement.

In this study, a circular-pull-down motion separation method is designed for apple picking based on a flexible three-finger end-effector (defined as separation method 3: shown in Figure 3c). After the gripper has held the apple, the apple is translated a certain distance so that the rootstock presents a certain angle to the vertical direction. The apple is moved in a circular motion with the gripper, while a pulling force is applied to the apple in the direction of the rootstock growth while keeping the angle constant. The circular movement of the apple and the increased tension in the direction of rootstock growth will cause the apple rootstock to break off and separate the apples and branches.

### 2.1.3. Force Analysis of Picking Pattern

As the apple moves in a circular motion, the branch pulls force $f_p$ is broken down into a vertical component $f_m$ and a horizontal component $f_n$, as shown in Figure 4. During fruit separation, the gripper acting on the apple growth direction vertical component force $f_{tm}$ gradually increases to a maximum value, a critical value for apple stem breakage, while after fruit separation is complete, the direction of $f_{tm}$ changes to the reverse growth direction, overcoming the effect of gravity and achieving a smooth grasping effect. The apple growth direction horizontal component force $f_{tn}$ overcomes the horizontal component pull force $f_n$, and provides the centripetal force $f_c$ required for the circular motion.

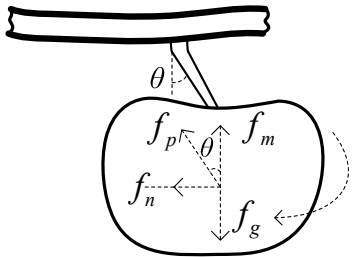

**Figure 4.** Force analysis of circular motion.

When picking apples using the inside finger horizontal grasp stance, the palm (cushioning substance) is in full contact with the apple, providing an upward thrust. The combined horizontal component of the medial finger pressure and palm thrust provides the centripetal force for the circular motion of the apple, and the vertical component drives the rhizome to break. The force state of the medial finger vertical grasp stance is similar to the medial finger horizontal grasp stance, as shown in Figure 5. Compared to the traditional grasp, the medial finger grasp has the advantage of increasing the contact area between the fingers and the apple, reducing the pressure on the skin of the apple and the area of damage to the apple after picking.

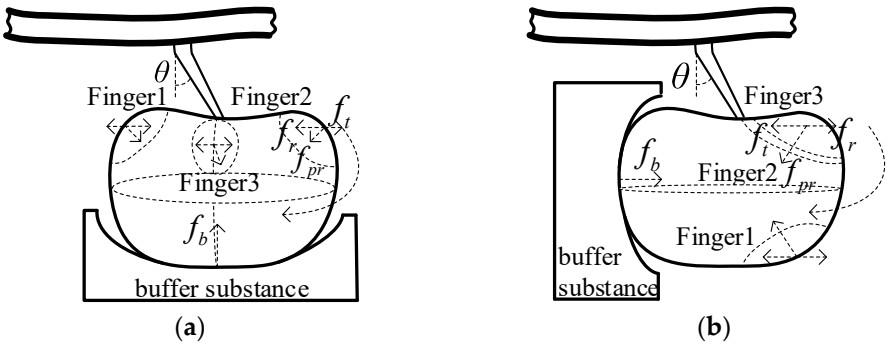

**Figure 5.** Force analysis of fingertip centripetal grasp and fingertip vertical grasp. (**a**) Fingertip centripetal grasp combined with circular-pull-down motion; (**b**) fingertip vertical grasp combined with circular-pull-down motion. Straight arrows indicate the force direction, curved arrows indicate the apple movement direction, and dotted curves indicate the apple cross-section and the part of the apple that is in contact with the finger.

*2.2. Dynamic Simulation*

Numerical simulations of multi-body systems are often used to assess their dynamical behavior due to the interference of the harvesting environment and measurement methods, as well as the complexity of the system [18,19]. Numerical simulations of multi-body systems are widely used for accurate modeling, mechanical analysis, and design optimization of different systems in agriculture [20,21]. In order to test the relationship between the kinematic principle of the circular-pull-down motion separation method and apple damage, a 3D branch–stem–apple model was built using SolidWorks (version 2018, Dassault System, Waltham, MA, USA), as shown in Figure 6. The material parameters are shown in Table 1.

Note that branches and stems are transversely orthotropic anisotropic materials [22,23]. Where $E_x$, $E_y$ and $E_z$ are the radial elastic modulus, tangential elastic modulus and axial elastic modulus, respectively. $G_{xz}$ is the radial bending shear modulus in the XZ plane, $G_{yz}$ is the radial bending shear modulus in the YZ plane, $G_{xy}$ is the axial torsional shear modulus, and u_mn is the Poisson's ratio in the mn corresponding plane. In addition, the mechanical properties of apples are defined by the elastic modulus (*E*) and Poisson's ratio (*u*) of the skin [24].

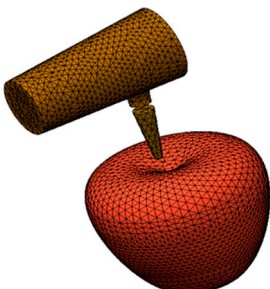

**Figure 6.** Finite element mesh model of branch, stem, and apple.

**Table 1.** Material parameters of the branch–stem–fruit model.

| | Density (kg m$^{-3}$) | E$_x$ (MPa) | E$_y$ (MPa) | E$_z$ (MPa) | u$_{xy}$ | u$_{xz}$ | u$_{yz}$ | G$_{xy}$ | G$_{xz}$ | G$_{yz}$ |
|---|---|---|---|---|---|---|---|---|---|---|
| Branch | 600 | 296 | 296 | 6274 | 0.49 | 0.063 | 0.063 | 33 | 310 | 310 |
| Stem | 300 | 29.8 | 29.8 | 439.8 | 0.49 | 0.031 | 0.031 | 26.7 | 1.8 | 1.8 |

| Apple | Density (kgm$^{-3}$) | E (MPa) | u |
|---|---|---|---|
| Skin | 840 | 12 | 0.35 |
| Cortex | 840 | 5 | 0.35 |
| Core | 950 | 7 | 0.35 |

In addition, a virtual simulation of apple circular motion was carried out using MSC.ADAMS for the study of the relationship between parameters such as the velocity of apple motion, the angle to the vertical and the root impulse during circular motion. Using the branch–stem–apple model to simulate the apple's circular motion during apple picking, vertical, horizontal, and rotational forces were applied to the apple, and a ball hinge connection was used between the stem and the apple, and the branch and the stem, as shown in Figure 7. In the simulation experiments, the horizontal and vertical forces play a decisive role in the apple's circular motion. The rotational force (0.015 N) was kept constant in value during the experiment, and the vertical and horizontal forces took values in the ranges 0 N to 70 N and 0 N to 5 N, respectively.

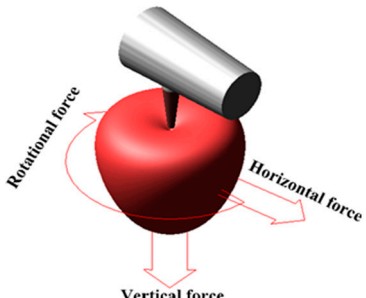

**Figure 7.** Dynamics simulation model of the branch–stem–apple in ADAMS.

### 2.2.1. Swing Angle and Motion Speed

During the apple circular motion, the horizontal thrust and vertical tension jointly determine the swing angle $\theta$ and the speed of the apple motion. When the horizontal thrust is excessively large (3.5 N~5 N) and the vertical pull is excessively small (0 N~3 N), the mean value of the angle $\theta$ oscillation (50°~65°) becomes large, and the apple movement is prone to instability. Figure 8 shows how the mean value of clamp angle $\theta$ swing is affected by horizontal thrust and vertical tension. The mean value of clamp angle $\theta$ swing shows a positive trend with horizontal thrust, while it shows an overall negative trend with vertical tension. A higher vertical pulling force ensures that the grasp angle $\theta$ oscillation is not too

large while at the same time satisfying the breaking force. For successful apple picking and to avoid instability, it should be ensured that the vertical pull force is higher than 5 N and the horizontal thrust force is stable in the range of 0.4 N~3 N. Equation (1) represents the corresponding regression equation between the mean clamp angle $\theta$ swing, the horizontal thrust and the vertical pull, where the value of $R^2$ is 0.9775. The maximum value of the angle $\theta$ swing is $1.5 \pm 0.52$ times the mean angle $\theta$ swing, as shown in Figure 9. Equation (2) represents the linear regression equation for the corresponding regression curve between the two, where the $R^2$ value is 0.9767.

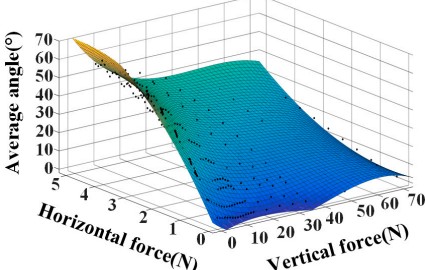

**Figure 8.** The relationship between vertical tension, horizontal thrust, and mean swing angle. The black points indicate the average angle values in the case of determining the vertical and horizontal force. As the average angle increases, the color of 3D graphics also shows a trend of change.

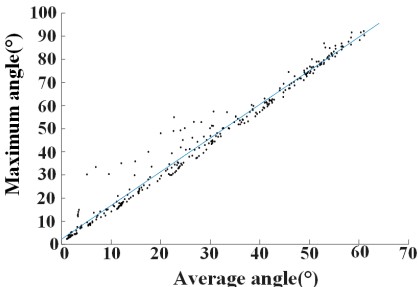

**Figure 9.** The relationship between the mean value and the maximum value of the swing angle. The black points indicate the maximum angle value, which corresponds to the average angle under the separation motion.

The speed of the apple movement and the horizontal thrust show a positive correlation; the horizontal thrust needs to be stabilized above 0.7 N to enable the apple to reach a stable stage of circular movement quickly. The vertical pull force is in the range of 18 N~55 N and has a good suppression effect on the speed of movement, which can limit the speed of movement to below 0.68 ms$^{-1}$. Figure 10 shows the average velocity of apple motion influenced by horizontal thrust and vertical pull. Equation (3) represents the corresponding regression equation between the mean speed of movement, the horizontal thrust, and the vertical pull, where the value of $R^2$ is 0.9816. When the mean speed of movement exceeds 1.5 ms$^{-1}$, the maximum speed deviation increases significantly, which is not conducive to the stability of the apple movement process, as shown in Figure 11. Equation (4) represents the linear regression equation for the corresponding regression curve between the two, where the value of $R^2$ is 0.9108.

$$\theta_{avg} = 3.787 - 1.071F_{ver} + 26.69F_{hor} + 0.0391F_{ver}^2 - 0.531F_{ver}F_{hor} - 2.537F_{hor}^2$$
$$-0.0003362F_{ver}^3 + 0.001965F_{ver}^2F_{hor} + 0.05424F_{ver}F_{hor}^2 \tag{1}$$

$$\theta_{Max} = 1.456\theta_{avg} + 2.273 \tag{2}$$

$$V_{avg} = -0.000816 - 0.009111F_{ver} + 0.5822F_{hor} + 0.0003814F_{ver}^2$$
$$-0.005366F_{ver}F_{hor} - 0.0305F_{hor}^2 - 0.0000029F_{ver}^3 \tag{3}$$
$$+0.0000037F_{ver}^2F_{hor} + 0.0007186F_{ver}F_{hor}^2$$

$$V_{Max} = 1.538V_{avg} + 0.07131 \tag{4}$$

where $\theta_{avg}$ and $\theta_{max}$ represent the mean and maximum values of the amplitude of oscillation of the angle $\theta$, $V_{avg}$ and $V_{max}$ represent the mean and maximum values of the speed of movement of the apple and $F_{ver}$ and $F_{hor}$ represent the vertical pull and horizontal thrust, respectively.

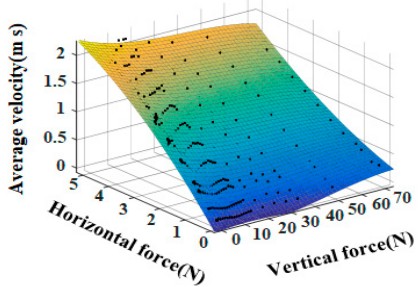

**Figure 10.** The relationship between vertical tension, horizontal thrust, and the average speed. The black points indicate the average velocity values in the case where the horizontal and vertical forces are determined. As the average velocity increases, the color of 3D graphics also shows a trend of change.

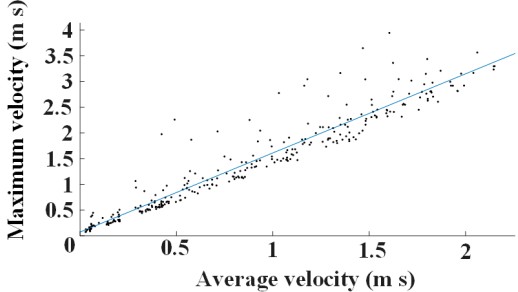

**Figure 11.** The relationship between the average and maximum velocity of apple movement. The black points indicate the maximum velocity value, which corresponds to the average velocity under the separation motion.

### 2.2.2. Picking Pattern Optimization

The degree of force on apple rootstocks during picking can be described in terms of impulse. The dynamic payload versus time curve derived from apple-picking experiments validates this idea well [14]. Apple rootstock impulse is governed by horizontal thrust, vertical tension, apple gravity, and rotational forces. Figure 12 shows the effect of horizontal thrust and vertical tension on apple root impulse, and Equation (5) represents the regression equation between several of these, where the value of $R^2$ is 0.9888, $I_r$ indicating the amount of impulse applied to the apple root.

$$I_r = 22.31 + 4.48F_{ver} + 13.91F_{hor} + 0.0291F_{ver}^2 - 0.595F_{ver}F_{hor} + 5.191$$
$$F_{hor}^2 - 0.000274F_{ver}^3 + 0.00475F_{ver}^2F_{hor} + 0.00507F_{ver}F_{hor}^2 \tag{5}$$

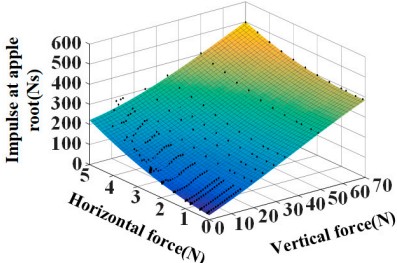

**Figure 12.** The relationship between horizontal tension, vertical thrust, and impulse.

The impulse volume has a positive trend with the horizontal thrust and the vertical pull. With a constant vertical pull (5 N), the horizontal thrust (1 N lifting to 3.5 N) can lift the impulse by 77.3 ± 14.1 N·s (80.2067 N·s), as shown in Figure 13 ($R^2$ = 1). With a constant horizontal thrust (1.5 N), a vertical pull (15 N to 50 N) can lift an impulse of 177 ± 9 N·s (173.64 N·s), as shown in Figure 14 ($R^2$ = 1). Compared to the horizontal thrust (0 N to 3.5 N), the apple's own gravity, and the rotational force (0.015 N), the vertical pull plays a dominant role in the impulse of the apple root in a circular motion.

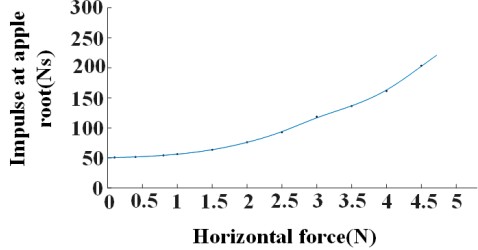

**Figure 13.** The relationship between horizontal thrust and impulse at constant vertical tension.

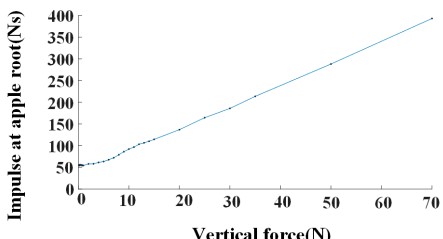

**Figure 14.** The relationship between vertical tension and impulse at constant horizontal thrust.

Neglecting the effect of the horizontal thrust, the impulse tends to be negatively correlated with the angle $\theta$. Figure 15 shows the relationship between impulse and angle $\theta$ when the horizontal thrust is at 1 N, 1.5 N, 2.0 N, 2.5 N, and 3.0 N, and the vertical pull varies from 0 N to 70 N. According to the experimental data of apple picking by [14], the value of the impulse required for successful apple picking was found to be 52 ± 16 N·s. On the premise that the impulse required for successful apple picking (70 N·s) and the horizontal thrust force was kept within the range of 1 ± 0.2 N, combined with Equations (1) and (3), the optimum angle $\theta$ corresponding to a vertical pull range of 4.62 N to 6.93 N and a circular motion of 17.6° to 22.8° was obtained.

### 2.3. Harvesting Robot System Overview

The harvesting robot developed is shown in Figure 16. It consists mainly of a mobile platform, a 5-degree-of-freedom manipulator, a flexible three-finger gripper, a stereo camera, a controller, and a host computer. When the camera finishes identifying and positioning the target apple [25], the host computer gives instructions to control the manipulator to align with the target apple and bring the gripper to a position where it can finish wrapping it [26].

Through the synergistic operation of the gripper and the robotic manipulator, picking the target apple in the desired grasp posture and separation is achieved.

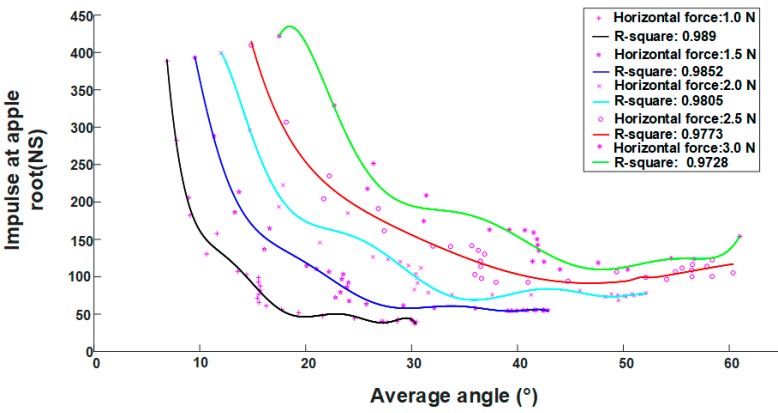

**Figure 15.** The relationship between the apple root impulse and the mean value of swing angle.

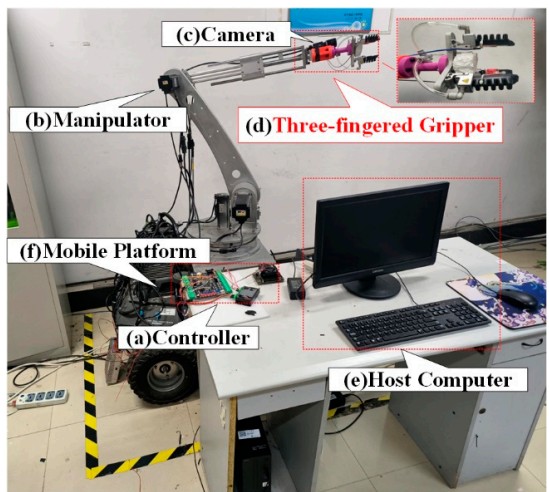

**Figure 16.** Apple harvesting robot.

For low damage, stable grasping and picking of apples, a three-fingered gripper is designed in this paper Figure 16. A cushioning substance is arranged in the inner hand position of the gripper to prevent unnecessary damage to the apples by colliding with the gripper hand position when grasping on the inside of the fingers. The controller controls the air pressure input to the three-finger gripper to complete the control of the degree of soft finger bending and the amount of inner contact force to achieve the grasping and release of the apple by the gripper. The upper computer controls the swing of the manipulator, and the controller controls the opening and contracting of the gripper to achieve different grasping postures of the gripper on the apple and complex separation movements of the apples and branches.

### 2.4. Gripper Control System

The gripper control system of the apple harvesting robot is shown in Figure 17 and consists of a three-fingered gripper, three thin film-type pressure sensors, a pneumatic pressure source system, an embedded controller, and an upper computer. Three thin film pressure sensors (Xinxin Microelectronics Technology Co., Ltd., Qingdao, China) are mounted on the inside of the soft finger of the gripper to obtain force data on the apple during the picking process and transmit the force data to the embedded controller (STM32F407 development board, MDK). The pneumatic pressure source system consists of an electrical proportional valve (SMC Corp., Tokyo, Japan), an air pressure regulator

(Chint Electric Co., Ltd., Yueqing, China), and an air compressor to provide air pressure. The air compressor provides a stable air pressure output, and the air pressure regulator and the electrical proportional valve ensure that the air pressure supply does not exceed the maximum capacity of the soft finger.

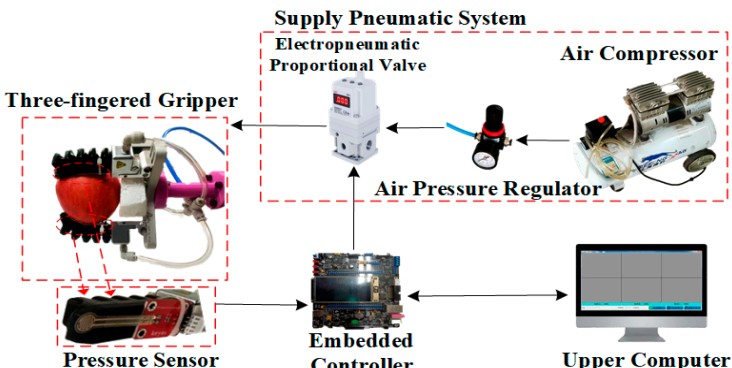

**Figure 17.** Apple harvesting robot gripper control system.

When the three-fingered gripper reaches the picking position, the embedded controller sends a voltage signal (0 V to 5 V) to the electrical proportional valve, which outputs air pressure (0 MPa to 0.9 MPa) in equal proportion. The pneumatic soft finger receives the air pressure and begins to bend inwards until the thin film-type pressure sensor (0 N to 20 N) outputs a force signal, and the gripper completes contact with the apple. As the input air pressure rises, the gripper moves from the contact state to the stable grasping state, and the corresponding apple-picking action is carried out according to the different separation methods. Once the apple picking is complete, the gripper returns to the grasping state. Figure 18 shows the four picking patterns tested in this paper. Throughout the picking process, the grasping force is detected using a thin-film type pressure sensor and fed back to the embedded controller. The embedded controller adjusts the input air pressure according to the force error until the apple is successfully picked.

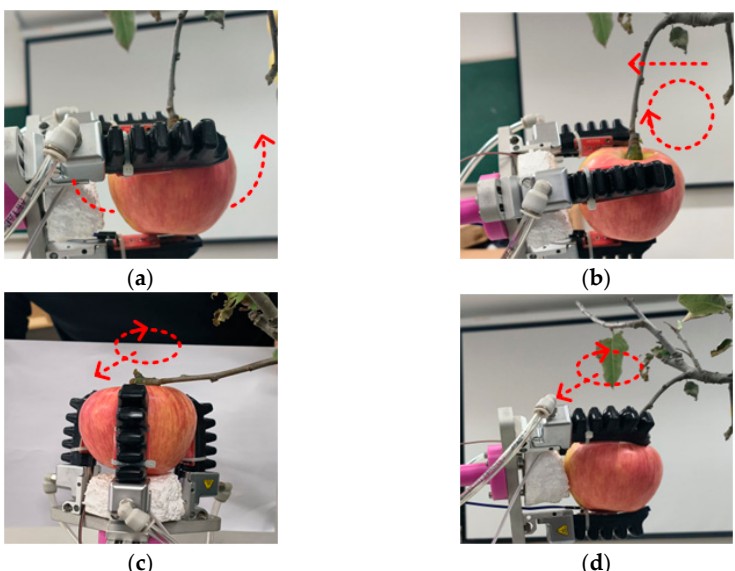

**Figure 18.** Apple picking patterns of a flexible three-fingered end-effector. (**a**) Picking pattern 1: fingertip bending–pulling motion pattern proposed in [13]; (**b**) picking pattern 2: fingertip rotating-horizontal pulling motion pattern proposed in [14]; (**c,d**) picking pattern 3/4-circular-pull-down motion pattern based on the inner finger grasp proposed in this study. (**a**) Picking pattern 1 [13], (**b**) picking pattern 2 [14], (**c**) picking pattern 3 [proposed], (**d**) picking pattern 4 [proposed]. The red arrow indicates the force direction.

In order to obtain the optimal swing angle and verify the superiority of the circular-pull-down separation picking pattern based on the flexible three-fingered end-effector, the optimal swing angle range experiment and optimal picking pattern experiment are designed in this paper. As far as possible, apples with the same growing environment, physical characteristics, and maturity were selected as experimental subjects. Thirty suitable target apples were selected for each picking pattern in the experiments. Acquire force data and analyze the degree of damage for the instances of successful picking among them.

### 3. Result and Discussion

*3.1. Optimal Swing Angle θ Range Experiment*

Figure 19 shows the dynamic curve of the gripper payload over time during apple picking using the three-fingered gripper in the circular-pull-down motion separation method. This separation method is demonstrated in Figure 18d. At the beginning of the picking process, the gripper interacts with the pulling force from the branch. As the picking process continues, the gripper starts to perform the picking action, and the force on the apple skin continues to increase until the apple is separated from the branch. With the angle $\theta$ at $17.5° \pm 2.5°$, the maximum payload reaches only $8.52 \pm 0.85$ N, which is better than the other ranges. This means that compared to conventional picking patterns, apple picking using the circular-pull-down motion separation method effectively reduces the forces on the apples during the picking process and reduces the amount of space needed for the end-effector to pick the apples. However, force fluctuations are more frequent, and the picking process consumes longer.

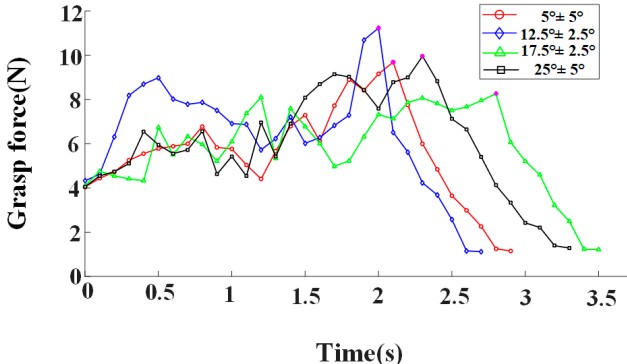

**Figure 19.** Dynamic changes in the payload of the gripper at different angles.

Table 2 shows the comparison of different ranges of angle $\theta$ in terms of grasping force, holding force, pressure, and picking time during the picking process. Ignoring the effects of factors such as modeled apple quality and the actual apple growing environment, the optimal range of grasping angle $\theta$ obtained from the picking experiments ($17.5° \pm 2.5°$) is largely consistent with the optimal range of angles ($17.6°$ to $22.8°$) obtained from previous dynamic simulation experiments. Within this range, the peak grasping pressure was $20.88 \pm 4.2$ KPa, and the grasp force was $1.24 \pm 0.41$ N.

**Table 2.** Parameters in different swing angle ranges.

| Swing Angle ($\theta$) | Average Peak Force (N) | Average Peak Pressure (KPa) | Average Hold Force (N) | Average Hold Pressure (KPa) | Average Pick Time (s) |
|---|---|---|---|---|---|
| $5° \pm 5°$ | 8.904708 | 21.825 | 1.380253 | 3.383 | 3.1 |
| $12.5° \pm 2.5°$ | 10.685652 | 26.19 | 1.089093 | 2.669 | 2.9 |
| $17.5° \pm 2.5°$ | 8.520288 | 20.883 | 1.243031 | 3.047 | 3.9 |
| $25° \pm 5°$ | 8.692691 | 21.306 | 1.457324 | 3.572 | 3.4 |

### 3.2. Optimal Picking Pattern Experiment

In this paper, comparative tests were designed for the optimal picking pattern. Table 3 shows the comparison of the different picking patterns in terms of grasping force, holding force, pressure, and picking time. Among them, patterns 1 and 2 (Figure 18a,b) are the optimal picking patterns designed in [13,14], respectively. Patterns 3 and 4 (Figure 16c,d) proposed in this paper are the circular-pull-down motion picking patterns within the optimal angle $\theta$ of the inner grasp stance. In contrast to the swinging amplitude of apples picked in the bending–pulling motion separation method, the circular-pull-down motion separation method is reduced by about 4°, decreasing the risk of collision between the target apple and adjacent apples during the picking process. In the grasping phase of pattern 3, the force on the skin of the apple is minimized by the pressure on the contact surface of the fingers, and the cushioning material in the palm position takes up the full weight of the apple, reducing the burden on the fingers. Comparing the four picking patterns experimentally, the three-finger gripper using picking pattern 4 outperforms the other picking patterns in terms of peak grasping force and pressure applied throughout the picking phase, but the picking takes the longest, as shown in Table 3. In the harvesting experiments using pattern 1 and pattern 2, it was found that there was a phenomenon of apples popping out due to excessive gripping force, and apple injury rates were 7% and 11%, respectively. Moreover, most of these injuries are due to excessive swinging into other objects and excessive force. In patterns 3 and 4, the apple is stably held by the end-effector without any sliding or falling off. Apple injury rates were 4% and 3%, respectively. During apple picking, appropriately increasing the speed of the apple stem separation movement and the pulling force along the direction of root growth can shorten the picking time required for pattern 4. Figure 20 shows the dynamic variation curves of payload with time for the three-finger gripper with the optimal picking pattern 4 designed in this paper and the optimal picking patterns designed by [13,14]. The finger medial grasp posture and the circular-pull-down motion of separation show potential for application in the apple-picking pattern of the apple harvesting robot.

**Table 3.** Parameters under different pickin12g patterns.

| Picking Pattern | Average Peak Force (N) | Average Peak Pressure (KPa) | Average Hold Force (N) | Average Hold Pressure (KPa) | Average Pick Time (s) |
|---|---|---|---|---|---|
| Pattern 1 [13] | 10.219213 | 60.113 | 2.834512 | 16.673 | 2.5 |
| Pattern 2 [14] | 14.713971 | 86.552 | 2.861234 | 16.831 | 1.9 |
| Pattern 3 [proposed] | 8.862305 | 21.721 | 0.452032 | 1.108 | 3.6 |
| Pattern 4 [proposed] | 8.520288 | 20.883 | 1.243031 | 3.047 | 3.9 |

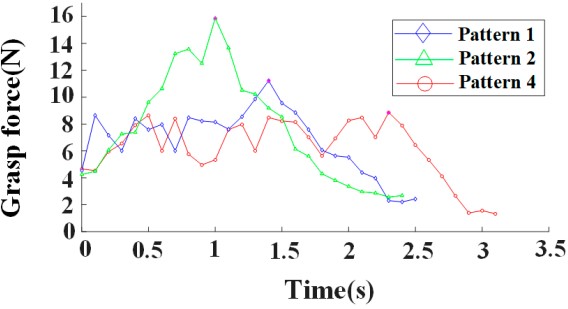

**Figure 20.** Dynamic changes in the payload of the gripper for different picking patterns.

### 4. Conclusions

In order to reduce the grasping force of apple picking by the harvesting robot, based on the existing picking patterns, a new picking pattern with a three-finger gripper circum-

ferential pull is proposed by designing the inner finger grip posture and circular-pull-down motion separation. Using the apple epidermal stress, apple motion speed, and apple root impulse as optimization parameters, the apple circular motion dynamics simulation experiments were carried out to determine the optimal range of angle θ (17.6°~22.8°) for the circular-pull-down motion picking pattern. By evaluating the effect of the simulated forces on the optimized parameters, the values of the simulated forces corresponding to the circular motion picking pattern are solved in the optimal angle range (horizontal thrust: $1 \pm 0.2$ N, vertical pull: 4.62 N~6.93 N). Finally, the apple harvesting robot experimental platform was built, and the picking performance of the four picking patterns was evaluated. The experimental results show that the circular-pull-down motion separation method has lower grasping forces than the curved-pull motion and the rotational-horizontal motion separation method. At the same time, it was determined that an inside-finger parallel stem-calyx axis grasping stance and a circular-pull-down motion separation were potentially the best combinations of picking patterns. This pattern had a peak average grasping force of 8.52 N during picking. In the future, additional research and experiments will be done on picking apples in harsh environments, such as gusty winds and torrential rains, using pattern 4.

**Author Contributions:** Conceptualization, W.J. and G.H.; methodology, B.X. and G.H.; software, G.H. and H.Z.; validation, H.Z.; formal analysis, G.H.; investigation, W.J.; data curation, G.H.; resources, W.J.; writing—original draft preparation, G.H.; writing—review and editing, W.J. and X.Y.; visualization, B.X.; supervision, W.J.; project administration, W.J.; funding acquisition, B.X. and W.J. All authors have read and agreed to the published version of the manuscript.

**Funding:** This research was funded by the National Natural Science Foundation of China (No. 61973141) and a project funded by the Priority Academic Program Development of Jiangsu Higher Education Institutions (No. PAPD).

**Institutional Review Board Statement:** Not applicable.

**Data Availability Statement:** Data are contained within the article.

**Conflicts of Interest:** The authors declare no conflict of interest.

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
