# Peer review of "A New Picking Pattern of a Flexible Three-Fingered End-Effector for Apple Harvesting Robot"

_agriculture, doi:10.3390/agriculture14010102_

Round 1
Reviewer 1 Report
Comments and Suggestions for Authors
This article studies the posture of the end effector grasping apples during the automated apple harvesting process, as well as the impact of the movement method of separating apples from the tree on the success rate of apple harvesting. The research idea is very clear, and the perspective and method of analyzing the problem are also good. It is indeed a serious study of the mechanical biological mechanical characteristics, which has a lot of reference significance for studying the efficiency and success rate of automated harvesting of circular fruits and vegetables such as apples. Overall, The article is written quite well. Here are a few suggestions:
1. The references are not particularly new, and there are relatively few articles in 2023. It is recommended that most of the literature use articles from the past 3 years, while a few use articles from within 5 years. Literature from 5 years or more should be used as little as possible, except for particularly classic articles. The citation position of references should be consistent, not at the beginning or end of the sentence, and a subject can be added appropriately. The name of a research unit or scholar should not appear in the text, which may not necessarily indicate proficiency in the content of the literature. Personal opinions should be considered by the author.
2. It is best for scientific research articles to have their own viewpoints, from beginning to end, throughout the entire text. The article is easy to understand and has strong readability. Do not constantly quote other people's viewpoints in key discussion paragraphs or analysis paragraphs. Instead, present your own opinions. Scientific research allows for mistakes and exploration. The inability to clearly see the author's viewpoint in the article is not conducive to readers understanding the article, and the readability of the article is not very good.
3. The problem to be solved in the research has not been clearly identified. Instead, a paragraph should be specifically listed to clearly identify the problem to be solved, the factors that affect it, and then the specific analysis process. The designed end effector is just a tool to verify the viewpoint of the article. It is not about creating an end effector. Once the data is produced, everything will be fine. There should be a clear viewpoint, an analysis process, comparative experimental verification, conclusions, and a summary discussion. The quality of the article will be better.
4. The analysis of the article should address the problem from a consistent perspective, such as improving picking efficiency and quality. However, there are still some differences. If the results obtained from both aspects are consistent, they can be analyzed in parallel. If analyzing from both perspectives can lead to a disorderly layout of the article, it is best to analyze from one perspective. Then, this analysis perspective should run from beginning to end, not just focusing on picking efficiency and quality, Suggest reading the entire text and considering some sentences.
5. Theoretical viewpoints are first proposed, then analyzed to obtain conclusions, followed by the design of end effectors and validation experiments. The logic of the article should be like this: the designed end effector should not be taken out first and placed before the theoretical analysis. Please adjust the relevant order.
6. Where the designed end effector corresponds to the theoretical part of the article should be clearly stated in the text for readers to understand. It is not enough to simply introduce the components of the designed end effector, and how each component corresponds one-to-one with the theory. It is also necessary to clearly explain. Which sensors were used to obtain what data, and which parts of the theoretical analysis corresponded one-to-one. Please organize this section into the paragraph introducing the end effector, so that readers can understand that the collected data is important data for verifying the viewpoint of the article. After data collection is completed, which formulas are used for data processing, and the physical meaning represented by the processed data is clearly stated. In this way, in the subsequent summary, it is evident that the processed data is clearly supporting the viewpoint of the article.
Overall, the idea of the article is quite good and practical, but the writing ideas are not very clear, the sense of hierarchy is not strong, and the information expression is not clear. I hope the author can summarize and adjust the order of the article, think about the viewpoints and information that the article needs to convey to readers, how to convey them, and achieve these goals. This is a great article.
Author Response
Many thanks for the valuable comments on the manuscript which are much appreciated. We have revised the paper according to these constructive comments. The detailed response to each comment is given below. The revised parts are all highlighted in red in revision paper. We wish our sincere efforts would satisfy the requirements.
- The references are not particularly new, and there are relatively few articles in 2023. It is recommended that most of the literature use articles from the past 3 years, while a few use articles from within 5 years. Literature from 5 years or more should be used as little as possible, except for particularly classic articles. The citation position of references should be consistent, not at the beginning or end of the sentence, and a subject can be added appropriately. The name of a research unit or scholar should not appear in the text, which may not necessarily indicate proficiency in the content of the literature. Personal opinions should be considered by the author.
Response and modification: Thank you for your comments. Following your suggestion, we have increased the citation count of articles in the past three years. At the same time, place the references uniformly at the end of the sentence. Removed the mention of research units in the article. (Please see page 9, Section 2.3, lines 288 and Section Reference for details in revision).
- It is best for scientific research articles to have their own viewpoints, from beginning to end, throughout the entire text. The article is easy to understand and has strong readability. Do not constantly quote other people's viewpoints in key discussion paragraphs or analysis paragraphs. Instead, present your own opinions. Scientific research allows for mistakes and exploration. The inability to clearly see the author's viewpoint in the article is not conducive to readers understanding the article, and the readability of the article is not very good.
Response and modification: Thank you for your comments. As far as this paper is concerned, the main objective is to propose new picking patterns based on the existing picking patterns with a three-finger type soft finger end-effector. (Please see page 1, Section Abstract, lines 12-17 for details). If only the picking patterns proposed in this paper are introduced without explaining other picking patterns. For the readers, it lacks the necessary comparison and the content is empty, so it is necessary to introduce the traditional picking patterns, and at the same time, it is convenient to design the comparison experiments with them in order to prove the superiority of the picking patterns proposed in this paper. Of course, considering the reviewer's suggestion of "clearly see the author's viewpoint", this paper points out the proposed method in the title of Figure to prevent readers from confusion. (Please see page 3, lines 120-121 and page 4, 147-150 for details in revision).
- The problem to be solved in the research has not been clearly identified. Instead, a paragraph should be specifically listed to clearly identify the problem to be solved, the factors that affect it, and then the specific analysis process. The designed end effector is just a tool to verify the viewpoint of the article. It is not about creating an end effector. Once the data is produced, everything will be fine. There should be a clear viewpoint, an analysis process, comparative experimental verification, conclusions, and a summary discussion. The quality of the article will be better.
Response and modification: Thank you for your comments. The purpose of this paper is to solve the problems of high pressure on apple skin and poor grasping stability in the process of traditional picking. In order to solve these two problems, starting with the holding posture and separation of fruit and stem, new picking patterns are proposed, and the feasibility of the picking pattern is verified by simulation. Finally, a comparative experiment with the traditional picking mode is designed to prove the superiority of the proposed picking mode. (Please see page 2, Section Introduction, lines 85-91 for details in revision).
- The analysis of the article should address the problem from a consistent perspective, such as improving picking efficiency and quality. However, there are still some differences. If the results obtained from both aspects are consistent, they can be analyzed in parallel. If analyzing from both perspectives can lead to a disorderly layout of the article, it is best to analyze from one perspective. Then, this analysis perspective should run from beginning to end, not just focusing on picking efficiency and quality. Suggest reading the entire text and considering some sentences.
Response and modification: Thank you for your comments. This article mainly addresses the two issues of stability and apple stress, proposing new grasping postures combined with a new separation method to form new picking patterns. Build a 3D branch-stem-apple model for dynamic simulation analysis, obtain conditions that can meet the stable operation of this picking pattern, (Please see page 6, lines 218-220 for details in revision), conduct actual picking experiments, obtain actual data, and verify whether the problems of unstable grasping process and excessive force are solved. Therefore, both the simulation and experimental parts of this article are aimed at ensuring the stability of the grasping process and reducing stress. In response to your question. This article adds a comparative description of the stability of apples during the experimental process. (Please see page 12, lines 379-385 for details in revision).
- Theoretical viewpoints are first proposed, then analyzed to obtain conclusions, followed by the design of end effectors and validation experiments. The logic of the article should be like this: the designed end effector should not be taken out first and placed before the theoretical analysis. Please adjust the relevant order.
Response and modification: Thank you for your comments. The Harvesting robot system overview has been moved to the M & M section. (Please see page 9, Section Harvesting robot system overview, lines 287-295 for details in revision).
- Where the designed end effector corresponds to the theoretical part of the article should be clearly stated in the text for readers to understand. It is not enough to simply introduce the components of the designed end effector, and how each component corresponds one-to-one with the theory. It is also necessary to clearly explain. Which sensors were used to obtain what data, and which parts of the theoretical analysis corresponded one-to-one. Please organize this section into the paragraph introducing the end effector, so that readers can understand that the collected data is important data for verifying the viewpoint of the article. After data collection is completed, which formulas are used for data processing, and the physical meaning represented by the processed data is clearly stated. In this way, in the subsequent summary, it is evident that the processed data is clearly supporting the viewpoint of the article.
Response and modification: Thank you for your comments. In the Optimal swing angle range experiment section, readers can use the angle range obtained from the experiment to solve the force data in the direction of the coordinate system for achieving circular motion of apples. The relevant content has been introduced in the dynamics simulation section. (Please see page 9, lines 280-284 and page 11, lines 357-359 for details in revision). In the comparative experiment of multiple picking patterns, the pressure data obtained from the pressure sensor reflects the stress state of the apple skin, and this article provides the stress data obtained from the experiments. (Please see page 12, Table 2 and page 13, Table 3 for details in revision).
Thank you again for your valuable suggestions and support on this article. We hope these replies can solve your doubts.
Reviewer 2 Report
Comments and Suggestions for Authors
- The research study evaluates different picking patterns of a flexible three-fingered end-effector fora possible apple harvesting robot.
- In the introduction, the authors have included a review about apple compression resistance. However, the results from these previous studies have not been considered in the present study (no relation of the previous studies about fruit damage have been considered in the procedure or results of the present study)
- No literature review has been addressed about fruit detachment. Considering that the gripper objective is grasping and detaching ( “holding the apple and the movement method of separating the apple”) it is crucial to review the several previous studies about forces related to detachment (fruit removal forces, torsion forces, fatigue…) and consider these previous studies for the simulation models.
https://doi.org/10.1016/j.compag.2021.106224https://doi.org/10.1016/0021-8634(69)90123-1
- In the M & M section there is no description about the actual fruit experimental design. It is crucial to compare and validate the simulation (2.3. Dynamic simulation, lines 187- and Table 1, line 199) using an experimental design using real fruit.
- Section 2.3.3. in M & M is confusing (it is not clear if it is a description of the methodology or it is a presentation of results).
- Lines 298-320 could be moved to M & M section (also Figure 15).
- Line 331- (“3.1. Optimal swing angle θ range experiment”) there are several studies with actual apples related to this experiment, this fact should be considered. Actual test with real fruit should be addressed.
- A discussion of the results should be included.
- There are not clear conclusions about the results. The comparison of the different patterns proposed has not been clearly presented. A clear conclusion about the most adequate pattern and the effect in a possible harvesting robot could be included.
- No relation with real fruit information has been addressed.
- No relation of the results with robot prototypes for apple harvesting has been presented. Besides, a brief review of the state of the art about apple harvesting could be useful to understand the necessity of robot harvesting.
- Lines 380-383 (“Using the apple epidermal stress, apple motion speed and apple root impulse as optimization parameters, the apple circular motion dynamics simulation experiments were carried out to determine the optimal range of angle θ (17.6°~22.8°) for the circular-pull-down motion picking pattern”) This conclusion does not have any significance if it is not related to actual fruit.
- The results need to be validated with an experiment with actual apples.
- It is crucial to add an experimental test (with actual fruit) to assess fruit damage while produced by the gripper (fingers) using the different patterns.
- It is crucial to add an experimental test (with actual fruit) to assess fruit removal force (and other forces) while the gripper is detaching the fruit.
- It is crucial to add an experimental test to prove that the gripper is not producing fruit damage. It could also be justified based on the results from previous studies about the maximum forces to produce fruit damage (with different apple varieties and different ripeness stages).
- Could the proposed gripper system and patterns be used with other type of fruits? Which limitations could be found?
Reviewer 3 Report
Comments and Suggestions for Authors
The article is devoted to the design of a three-finger gripper as a working part of robots for harvesting apples, as well as to the study of the physical properties of the impact of the gripper on various parts of the apple.
The article is of scientific and practical interest and corresponds to the topics of the journal “Agriculture”. The article is written in clear language, contains technical design details, mathematical description and test results.
To publish in the journal, authors must make some adjustments:
1) In section 1, as well as 2.1, 2.2, it should be indicated which scientists in different countries dealt with this problem. It might even be worth adding a small “Literature Review” section. For example, “Belgian researchers M. Van Zeebroeck et al. investigated the mechanical properties of apple damage [13].”
2) At the end of the first section, it is advisable to clearly formulate the purpose of the study and tasks in points: 1,2,3...
3) Also, the conclusions should have been formulated more clearly in accordance with the objectives. Those conclusions that are written are more suitable for the “Discussion” section (it can also be separated into a separate section)
4) The methodology of the article can be presented graphically in the form of a picture using mind maps, or I would recommend making a graphic annotation.
5) Please pay attention to the practical significance of your research. What do the graphs and figures you received show in practice? How will this affect the safety of the fetus, how much will the % of damage decrease, etc.? What is the expected economic impact?
6) It would be nice to demonstrate a video of the capture in action; you could make a link to the video and include it in the article.
7) When describing the control scheme, add a paragraph about the software that is used for control. What type of controller is used? What programming language? It would be nice to take a screenshot.
In general, the article is serious and useful, I thank the authors for the work done.
Reviewer 4 Report
Comments and Suggestions for Authors
The article provides a clear and logical flow of information, starting from the improved three-finger fingertip grasp posture and separating method, followed by computational dynamic simulation experiments and the apple harvesting robot experimental platform was built and the picking performance of the four picking patterns was evaluated.
The findings show that the inside-finger parallel stem-calyx axis grasping stance and a circular-pull-down motion separation were potentially the best combination of picking patterns. The study contributes to the field of picking robot arm and it lays a foundation for the further research of the subsequent apple picking robot arm.
But here are some comments and suggestions to strengthen the article:
① Clarify the novelty: Provide a more detailed explanation in the introduction about the specific novelty or contribution of the research. Clearly state how the proposed approach or findings differ from existing methods or studies in the field of apple picking robot arm.
② Provide more picture: In view of the description in 2.2.3, can Figure 4 be supplemented and completed? The force analysis of the entire picking process of apple picking should be drawn clearly by stages.
③ Present and discuss results in more detail: Provide a thorough analysis and interpretation of the obtained results. Discuss any unexpected findings or limitations encountered during the study. Compare the results with existing literature and provide possible explanations for any discrepancies. For example, for the analysis in Table 2, it is not clear what evaluation criteria are used to determine as the optimal picking Angle range. If we use the average picking time as an evaluation criterion, it is clear that is better than other angles.
④ Address potential limitations: Identify and address any limitations or assumptions made in the study. Discuss the potential impact of these limitations on the validity and generalizability of the results.
⑤ Discuss practical implications: Provide a discussion on the practical implications of the research findings. How can the results be applied in real-world scenarios? Is the pneumatic structure suitable for outdoor platforms, and how to solve this problem in the future. Highlight the potential benefits and practical applications of the proposed approach.
⑥ Conclusion and future directions: Summarize the key findings and their implications in the conclusion section. Additionally, provide suggestions for future research directions to further advance the field or address any remaining gaps in knowledge.
Comments on the Quality of English LanguageEnglish very difficult to understand/incomprehensible
Author Response
Many thanks for the valuable comments on the manuscript which are much appreciated. We have revised the paper according to these constructive comments. The detailed response to each comment is given below. The revised parts are all highlighted in grey in revision paper. We wish our sincere efforts would satisfy the requirements.
- Clarify the novelty: Provide a more detailed explanation in the introduction about the specific novelty or contribution of the research. Clearly state how the proposed approach or findings differ from existing methods or studies in the field of apple picking robot arm.
Response and modification: Thank you for your comments. Relevant content has been amended. (Please see page 1, Section Abstract, lines 11-12 for details in revision).
- Provide more picture: In view of the description in 2.2.3, can Figure 4 be supplemented and completed? The force analysis of the entire picking process of apple picking should be drawn clearly by stages.
Response and modification: Thank you for your comments. Added Figure 4 to the relevant descriptions. (Please see page 5, Figure 4, lines 174-176 for details in revision).
- Present and discuss results in more detail: Provide a thorough analysis and interpretation of the obtained results. Discuss any unexpected findings or limitations encountered during the study. Compare the results with existing literature and provide possible explanations for any discrepancies. For example, for the analysis in Table 2, it is not clear what evaluation criteria are used to determine as the optimal picking Angle range. If we use the average picking time as an evaluation criterion, it is clear that is better than other angles.
Response and modification: Thank you for your comments. The main objective of this paper is to achieve the goal of reducing the damage rate by reducing the peak force on the apple during the gripping process, so the criterion of the study is that kind of pattern has the least peak force. Due to our mistake, we did not add the conclusion related to damage and stability, and apologize for causing you to have such a misunderstanding. The relevant content has been added. (Please see page 12, lines 376-385 for details in revision).
- Address potential limitations: Identify and address any limitations or assumptions made in the study. Discuss the potential impact of these limitations on the validity and generalizability of the results.
Response and modification: Thank you for your comments. The picking model proposed in this paper improves stability and reduces the force required for gripping, but consumes a longer picking time. For the picking robot, the picking operation can be carried out throughout the day. It does not happen that the harvesting operation cannot be completed during the apple harvesting time. Also, the article provides a way to reduce the harvesting time by slightly decreasing the stability. (Please see page 12, lines 385-387 for details in revision).
- Discuss practical implications: Provide a discussion on the practical implications of the research findings. How can the results be applied in real-world scenarios? Is the pneumatic structure suitable for outdoor platforms, and how to solve this problem in the future. Highlight the potential benefits and practical applications of the proposed approach.
Response and modification: Thank you for your comments. We can understand your concerns. First of all, compared to an electric end-effector, a pneumatic structure requires an air compressor to provide air pressure, which results in the apple picking robot needing a larger space to accommodate the corresponding air supply system. However, apple picking is not particularly dependent on large air pressure, and a small air compressor can be designed to solve the space problem. Secondly, pneumatic structures are more prone to damage than electric structures, especially hoses. However, in the orchard, collisions with common tree branches and leaves will not cause damage to the hose. Therefore, the robot introduced in this paper is feasible to be utilized in actual outdoor for picking operations.
- Conclusion and future directions: Summarize the key findings and their implications in the conclusion section. Additionally, provide suggestions for future research directions to further advance the field or address any remaining gaps in knowledge.
Response and modification: Thank you for your comments. In the future, we will optimize the mechanics of the apple robot to reduce the time required for picking. At the same time, we will conduct further research and discussions on the difficulties in recognizing apples and how to ensure smooth picking in bad weather, such as windy and rainy weather. Relevant content has been modified. (Please see page 13, lines 412-414 for details in revision).
Thank you again for recognizing this article and for your valuable suggestions, and we hope that these responses address your questions.
Round 2
Reviewer 2 Report
Comments and Suggestions for Authors
The authors have answered to all the questions presented by the reviewer. However, there is still some questions about the fruit validation test that could be clarified.
12. The results need to be validated with an experiment with actual apples. - It is crucial to add an experimental test (with actual fruit) to assess fruit damage while produced by the gripper (fingers) using the different patterns.
- It is crucial to add an experimental test (with actual fruit) to assess fruit removal force (and other forces) while the gripper is detaching the fruit. - It is crucial to add an experimental test to prove that the gripper is not producing fruit damage. It could also be justified based on the results from previous studies about the maximum forces to produce fruit damage (with different apple varieties and different ripeness stages).
Response and modification: Thank you for your comment.
First, injury data for different picking patterns have been added to the section on experiments with different picking patterns. (Please see page 12, lines 379-385 for details).
➔
The experimental design of this injury test should be explained (factors, variables, repetitions, procedure).
Secondly, if by "fruit removal force" you mean the breaking force of the Apple-Stem, then it has actually been shown in Figure 18 and 19. The moment of the peak of the curve in the picture is the moment when the breaking force is generated, which is three times (number of fingers) the peak force.
➔
The moment of the peak of the curve could be clearly defined in the graph. However, I understand that these curves are not based on measurement with real fruits. If it is so, this experiment should be defined (factors, variables, repetitions, procedure)..
Besides, the rupture of the apple-stem (known as removal force or detachment force) could be due to a maximum force or to fatigue (by repetitive deterioration of the tissue). This fact should be considered and could be different depending on the grasping pattern of the gripper.
Finally, data related to damage rates have been added, and experimental validation of apple picking experiments for different varieties with different ripening periods will be carried out in future studies.
➔
This information (“data related to damage rates have been added”) should indicated (line number).
Author Response
Thank you very much for your reply and valuable comments on the manuscript. We have revised the paper based on these constructive comments. The detailed response to each comment is given below. The revised parts are all highlighted in Green in revision paper. We wish our sincere efforts would satisfy the requirements.
- The experimental design of this injury test should be explained (factors, variables, repetitions, procedure).
Response and modification: Thank you for your comments. In the actual grasping experiments, we will first inspect the target apple (Regarding the selection criteria of target apples and the number of experiments, please see page 11, lines 336-342 for details in revision) to make sure that there are no scuffed and damaged parts on the surface of the apple. During the experiment, when the end-effector touches the apple, we will mark the contact position immediately. After picking, we checked the skin of the apples for obvious abrasions, and placed the apples in a low-temperature environment for a period of time to check whether there was any discoloration of the marked contact locations. If the skin of the apple is bruised or discolored, then it has been damaged.
2.The moment of the peak of the curve could be clearly defined in the graph. However, I understand that these curves are not based on measurement with real fruits. If it is so, this experiment should be defined (factors, variables, repetitions, procedure). Besides, the rupture of the apple-stem (known as removal force or detachment force) could be due to a maximum force or to fatigue (by repetitive deterioration of the tissue). This fact should be considered and could be different depending on the grasping pattern of the gripper.
Response and modification: Thank you for your comments. The data for our experiments came from real apple picking experiments. In the picking experiment, the output of force data was performed at the same time as the picking, so the observer was able to confirm that the time of complete breakage of the apple rootstock corresponded to the moment of peak force. At this stage of our research, we have also taken into account your concern about the "repetitive deterioration of the tissue". Combined with the experimental data from the picking pattern of Li, Fan et al. we tend to favor the idea that the constant application of force during gripping results in repetitive deterioration of the stem tissue, which in turn leads to the harvesting of the apples. A picking pattern with a short picking time puts more force on the apples, while a picking pattern with a long picking time puts less force on the apples in comparison. The long picking process causes repetitive deterioration of the stems, causing them to break. We would like to thank you for your valuable proposal and will continue to study this aspect in the future.
- This information (“data related to damage rates have been added”) should indicated (line number).
Response and modification: Thank you for your comments. Please see page 13, lines 392-398 for details in revision.
Finally, thank you again for your constructive suggestions on this article, and we hope that the responses above address your questions.
Reviewer 3 Report
Comments and Suggestions for Authors
In this version of the article, the authors tried to take into account my comments.
In particular, information about researchers working on this problem in sections 2.1 and 2.2 has been added, the goal has been more clearly formulated, and the conclusions have been shortened. A paragraph has also been added confirming the economic feasibility of the study in numbers.
Now I understand that the economic effect is associated with a decrease in apple injuries.
The hardware and software of the robotic system are described.
I also liked the new drawings on the structure of research in the form of mind maps, as well as the board and interface screen. However, for some reason they are presented only in the reviewer’s response, and are not presented in the main version of the article.
I would still recommend including them in the final version of the article (possibly in the Appendix - at the discretion of the editors).
It is also advisable to enlarge the figures with graphs, align the fonts in the figures, move the line labels to the right (in Fig. 14) so that they do not merge.
Author Response
Thank you very much for your reply and valuable comments on the manuscript. We have revised the paper based on these constructive comments. Detailed responses to each comment are given below. The revised parts are marked in Blue. We hope that our sincere efforts will fulfill the requirements.
- I also liked the new drawings on the structure of research in the form of mind maps, as well as the board and interface screen. However, for some reason they are presented only in the reviewer’s response, and are not presented in the main version of the article. I would still recommend including them in the final version of the article (possibly in the Appendix - at the discretion of the editors).
Response and modification: Thank you very much for your reply and recognition of the mind map we provided. Due to our negligence, we forgot to add it in the main text. We apologize for this. The revised mind map has been added, which can indeed help readers to clarify the structure of the article. (Please see page 3, lines 94, Figure 1 for details in revision).
- It is also advisable to enlarge the figures with graphs, align the fonts in the figures, move the line labels to the right (in Fig. 14) so that they do not merge.
Response and modification: Thank you for your comments. The relevant images have been corrected. (Please see page 9, lines 289, Figure 15 for details in revision).
Finally, thank you once again for your careful reading of this article as well as your valuable suggestions, and we hope the revised article will fulfill your requirements.
Reviewer 4 Report
Comments and Suggestions for Authors
I would like to congratulate the authors for their involvement in considering the reviewers suggestions and recommendations.
Comments on the Quality of English Language
Minor editing of English language required
Author Response
Thank you very much for your reply and valuable comments on the manuscript. We have revised the paper based on these constructive comments. Detailed responses to each comment are given below. The revised parts are marked in Grey. We hope that our sincere efforts will fulfill the requirements.
1. Minor editing of English language required
Response and modification: Thank you very much for your careful reading of this article, for the English language problems in this article, we have recorrected them to make the language of the article more concise. (Please see page 1, lines 15-16, 21-22, 25-26, 44-45; page 2, lines 49-50; page 3, lines 102-103; page 4, lines 142-143; page 6, lines 203, 206-207, 213 for details in revision).
Thank you again for your support and recognition of this article, as well as your valuable suggestions are also of great value and significance to the revision of our article.